# Curcumin Improved Glucose Intolerance, Renal Injury, and Nonalcoholic Fatty Liver Disease and Decreased Chromium Loss through Urine in Obese Mice

Geng-Ruei Chang [1],[†] , Wen-Tsong Hsieh [2],[3],[4],[†] , Lan-Szu Chou [5], Chen-Si Lin [6] , Ching-Fen Wu [1], Jen-Wei Lin [7], Wei-Li Lin [7],[8], Tzu-Chun Lin [1], Huei-Jyuan Liao [1], Chen-Yung Kao [1] and Chuen-Fu Lin [9],*

[1]  Department of Veterinary Medicine, National Chiayi University, 580 Xinmin Road, Chiayi 60054, Taiwan; grchang@mail.ncyu.edu.tw (G.-R.C.); cfwu@mail.ncyu.edu.tw (C.-F.W.); lin890090@gmail.com (T.-C.L.); pipi324615@gmail.com (H.-J.L.); wg141319@gmail.com (C.-Y.K.)
[2]  Department of Pharmacology, China Medical University, 91 Hsueh-Shih Road, Taichung 404333, Taiwan; wthsieh@mail.cmu.edu.tw
[3]  Chinese Medicine Research Center, China Medical University, 91 Hsueh-Shih Road, Taichung 404333, Taiwan
[4]  Drug Development Center, China Medical University, 91 Hsueh-Shih Road, Taichung 404333, Taiwan
[5]  Department of BioAgricultural Sciences, National Chiayi University, 300 Syuefu Road, Chiayi 60004, Taiwan; choulb@mail.ncyu.edu.tw
[6]  School of Veterinary Medicine, National Taiwan University, 4 Section, 1 Roosevelt Road, Taipei 10617, Taiwan; cslin100@ntu.edu.tw
[7]  Bachelor Degree Program in Animal Healthcare, Hungkuang University, 6 Section, 1018 Taiwan Boulevard, Shalu District, Taichung 433304, Taiwan; jenweilin@hk.edu.tw (J.-W.L.); ivorylily99@gmail.com (W.-L.L.)
[8]  General Education Center, Chaoyang University of Technology, 168 Jifeng Eastern Road, Taichung 413310, Taiwan
[9]  Department of Veterinary Medicine, College of Veterinary Medicine, National Pingtung University of Science and Technology, 1, Shuefu Road, Neipu, Pingtung 912301, Taiwan
*  Correspondence: cflin2283@mail.npust.edu.tw; Tel.: +886-8-7703202
†  These authors contributed equally to this work.

**Abstract:** Obesity-associated hyperglycemia underlies insulin resistance, glucose intolerance, and related metabolic disorders including type 2 diabetes, renal damage, and nonalcoholic fatty liver disease. Turmeric root is commonly used in Asia, and curcumin, one of its pharmacological components, can play a role in preventing and treating certain chronic physiological disorders. Accordingly, this study examined how high-fat diet (HFD)-induced hyperglycemia and hyperlipidemia are reduced by curcumin through changes in fatty liver scores, chromium distribution, and renal injury in mice. Relative to the control group, also fed an HFD, the curcumin group weighed less and had smaller adipocytes; it also had lower daily food efficiency, blood urea nitrogen and creatinine levels, serum alanine aminotransferase and aspartate aminotransferase levels, serum and hepatic triglyceride levels, and hepatic lipid regulation marker expression. The curcumin-treated obese group exhibited significantly lower fasting blood glucose, was less glucose intolerant, had higher Akt phosphorylation and glucose transporter 4 (GLUT4) expression, and had greater serum insulin levels. Moreover, the group showed renal damage with lower TNF-α expression along with more numerous renal antioxidative enzymes that included superoxide dismutase, glutathione peroxidase, and catalase. The liver histology of the curcumin-treated obese mice showed superior lipid infiltration and fewer FASN and PNPLA3 proteins in comparison with the control mice. Curcumin contributed to creating a positive chromium balance by decreasing the amount of chromium lost through urine, leading to the chromium mobilization needed to mitigate hyperglycemia. Thus, the results suggest that curcumin prevents HFD-induced glucose intolerance, kidney injury, and nonalcoholic fatty liver disease.

**Keywords:** curcumin; chromium; fatty liver; glucose intolerance; renal injury; obesity

## 1. Introduction

Diabetes mellitus (DM)—which can be of type 1, type 2, and gestational—is a crucial health concern in developing countries [1,2]. The global prevalence of type 2 DM has increased along with that of obesity. Obesity is associated with altered dietary habits; decreased physical activity and longevity; and impaired action of insulin on muscular, adipose, and hepatic tissues. In patients with obesity, regular exercise and a reduction in energy intake may lead to adequate glycemic control, followed by reductions in the likelihoods of microvascular complications, morbidity, and mortality [3]. However, individualized treatment targets are not possible for numerous patients with obesity. If, in response to increased insulin demand, pancreatic β-cells do not secrete sufficient insulin, severe hyperglycemia develops [4]. Currently, insulin therapy is considered essential because the underlying gradual decline of pancreatic β-cell function is directly linked to glycemic control deterioration. When glycemic targets are not achieved through treatment with hypoglycemic agents other than insulin, insulin therapy more satisfactorily manages DM [5,6]. Neuropathy, nephropathy, and retinopathy are among the microvascular complications that become less likely to occur when insulin therapy (and the subsequent insulin intensification) is employed. Various insulin therapies are available for DM management and achieve satisfactory glycemic control (e.g., multiple daily insulin injections or insulin pumps for type 1 DM, long-acting analogs and oral antihyperglycemic agents for type 2 DM, and a specific insulin type for gestational DM) [5]. However, the long-term administration of insulin can affect tolerance and lifestyle and increase the risks of hypoglycemia and weight change [7].

Turmeric root (Curcuma longa rhizome), used widely in cooking in Asia, contains the pharmacological component 1,7-bis(4-hydroxy-3-methoxy phenol)-1,6-heptadiene-3,5-dione (curcumin), a lipophilic polyphenol. Curcumin's antiaging, antioxidative, antibiotic, anti-inflammatory, and anticancer effects have been indicated by previously executed clinical trials and animal studies [8,9]. To date, no human study has reported adverse side effects of curcumin administration, except for itching, constipation, or vertigo in a few participants [10]. In some recent experiments executed in vivo and in vitro, curcumin was discovered to increase the activities of antioxidant enzymes, decrease the degree of hyperglycemia, correct lipid metabolism disorders, and decrease the degree of insulin resistance [11,12]. However, as indicated by some clinical studies, administering curcumin engendered no changes in fasting glucose level, body weight, an insulin resistance index (homeostatic model assessment of insulin resistance, HOMA-IR), or triglyceride levels [13–15]. Moreover, curcumin was identified to not affect insulin's stimulatory action in Akt phosphorylation within insulin signaling; hence, curcumin was concluded to directly inhibit basal and insulin-stimulated glucose uptake through GLUT4 [16]. Curcumin activated autophagy, inhibited the function of the ubiquitin–proteasome system, and reduced the Akt content of 3T3-L1 adipocytes; these actions were accompanied by the attenuation of glucose uptake that leads to resistance to insulin [17]. In one clinical trial, relative to lifestyle modifications alone, curcumin supplementation combined with lifestyle modifications did not display a superior effect in reducing inflammation within nonalcoholic fatty liver disease (NAFLD) [18]. By contrast, curcumin supplements significantly reduced the fat content of the liver in a study collecting ultrasonographic evidence of NAFLD [19].

Trace elements affect the regulation of protein, lipid, and carbohydrate metabolism. Chromium in particular is necessary for this metabolism and can have positive effects in individuals with neuropathy, glucose intolerance, DM, or obesity [6]. The clinical study of Chang et al. [20] documented the benefits of chromium to type 2 DM and insulin resistance. Furthermore, chromium made the downstream effectors of insulin signaling (e.g., Akt) more active in the study executed by Hua et al. [21]; this eventually enhanced cellular glucose uptake regulation and the translocation of GLUT4 to cell surfaces. However, chromium deficiency may promote type 2 DM and obesity by affecting normal glucose tolerance and lipid profile maintenance [3]. In addition, chromium was shown in vitro in human hepatoma SMMC-7721 cells to alleviate steatosis and help resist damage to the

liver through reduced synthesis of triglycerides and uptake of fatty acids [22]. Chromium supplementation was demonstrated by Chen et al. to improve the hallmarks of liver damage within a chronic cholestasis rat model [23]. Chromium has hypoglycemic, antioxidant, hypolipidemic, and anti-inflammatory properties; it can also potentially reduce the likelihood of complications in NAFLD [24]. Therefore, the present study hypothesized that hepatic lipid accumulation would be affected by curcumin because this substance alters the modulation of hepatic chromium levels, potentially affecting the treatment of NAFLD.

Curcumin's influences on the homeostasis of glucose in peripheral tissues and the corresponding insulin-linked mechanisms are equally worthy of investigation. Considering the dual influences on metabolic syndrome, curcumin's currently diverse reported effects on hyperglycemia pathogenesis can be attributed to the differing degrees to which it retards metabolic syndrome or its complete failure to do so. Thus, given the inconsistent findings on chromium's properties and the insufficient number of human and animal studies on chromium levels after curcumin supplementation, this trial study examined chromium's effects on blood glucose indices, lipid profiles in hyperglycemia, renal impairment, and NAFLD. More specifically, this research probed whether subjecting obese animals with hyperglycemia to incessant curcumin treatment would influence the animals' weight, expression of insulin signaling proteins, lipid levels, blood glucose levels, fatty liver status, kidney function, and endocrine profile. Renal failure influences lipid metabolism and more adversely affects people with obesity and DM or glucose intolerance [25–27]. Additionally, this study administered glucose tolerance tests (GTTs) and examined pancreatic β-cell function and insulin resistance (IR) to identify curcumin's effects on glucose homeostasis and mechanisms involving insulin. Such techniques are appropriate means of analysis and have been widely adopted in research on DM.

## 2. Materials and Methods

### 2.1. Mice

The National Laboratory Animal Center's Education Research Resource (Taipei, Taiwan) was the source of the current study's 5-week-old male C57BL/6J mice. The Taiwanese government stipulates that all animal experiments must adhere to the Guide for the Care and Use of Laboratory Animals; hence, the study methods were in accordance with this guide and were reviewed and approved by the National Chiayi University Institutional Animal Care and Use Committee (IACUC Approval No. 107029). Euthanasia of the mice was performed using carbon dioxide combined with 1.2 mg/kg urethane (an anesthetic overdose).

### 2.2. Experiment

The individual microisolation cages in which the mice were held were placed on a high-efficiency particulate air-filtered ventilated rack (Rungshin IVC Systems, Taichung, Taiwan) in the institution's animal quarters, which had a 12 h light–dark cycle and were maintained at 22 ± 1 °C and 55% ± 5% humidity. During the two-week acclimatization period, a standard diet (SD; 3.3 kcal/g metabolizable energy; diet 5008; PMI Nutrition International, Brentwood, MO, USA) was given ad libitum to the animals. This was then replaced by a high-fat diet (HFD, 5.16 kcal/g metabolizable energy; diet 58Y1, modified laboratory w/31.66% lard; 1.12 µg/g chromium; PMI Nutrition International), administered ad libitum for 10 weeks. This is in contrast to the usual four-week duration adopted by other scholars to establish obesity [6]. Once obesity and glucose intolerance had been induced, some of the then 17-week-old mice were selected and divided into control and curcumin subgroups (n = 10 each; average body weight of control vs. curcumin group: 36.37 ± 0.52 vs. 36.51 ± 0.15 g; no significant difference). The curcumin group was administered curcumin (100 mg/kg in 0.5% carboxy methyl cellulose freshly prepared and then delivered within 15 min) through daily oral gavage for 10 weeks, whereas the control-group mice were given gavage of the vehicle alone. The curcumin dose was based on previous research in which curcumin was administered to treat oxidative stress, hyperten-

sion, insulin resistance, and endometriosis in mice [28–31]. The mice were weighed on a weekly basis. After nine weeks of oral gavage with curcumin or vehicle, an intraperitoneal GTT (IPGTT) was performed to estimate glucose tolerance. Before being euthanized, the animals were held in their own metabolic cage for 12 h, during which samples of urine were obtained from the cage. They had ad libitum water access during the 12 h. The mice were euthanized once the experiment was complete. Their gastrocnemius muscle, liver, kidney, retroperitoneal fat pad, epididymal fat pad, and femoral bone tissue were harvested. In addition, whole blood and serum were collected to determine hormone and chromium levels and conduct a molecular protein analysis.

### 2.3. Body Weight, Insulin Level, and Food Intake

The mice's food consumption and body weight was measured weekly. The remnant food in each cage dispenser and that on the cage's floor was weighed to calculate food intake [25]. To measure serum insulin levels, blood samples were analyzed using mouse insulin (#INSKR020) enzyme-linked immunosorbent assay kits (Crystal Chem, Downers Grove, IL, USA).

### 2.4. Histology and Morphometry of Liver, Fat Pads, and Kidneys

The mass of the harvested livers, epididymal white adipose tissue (EWAT), and retroperitoneal white adipose tissue (RWAT) was measured in grams. Hematoxylin and eosin (H&E) staining was executed to determine the extent of fat infiltration into the liver. Each tissue was assigned a score of 0, 1, 2, 3, or 4 when 0%, <5%, 5–25%, 25–50%, or >50% fat infiltration was visualized, respectively [10]. For the epididymal fat pads, adipocyte size was systematically analyzed for numerous tissue sections. Each tissue section was stained with H&E, and at least 10 fields (~100 adipocytes) were assessed per section [25,26]. Each kidney was bisected along its longitudinal axis with a long sharp blade and then stained with H&E. Researchers blinded to the origin of the specimens evaluated whether glomerulonephritis was present [25]. All imaging was performed with a digital microscope with a high resolution (Moticam 2300; Motic Instruments, Richmond, BC, Canada). For the determination of the adipocyte size distribution, this study executed Motic Images Plus (v. 2.0, Motic China Group Co., Ltd., Xiamen, China, 1999–2004) [26].

### 2.5. Serum and Hepatic Triglyceride Levels and Serum Alanine Aminotransferase and Aspartate Aminotransferase Levels

After euthanization, samples of blood were derived from all of the mice. The serum triglyceride, serum alanine aminotransferase (ALT), and serum aspartate aminotransferase (AST) concentrations of the blood samples were determined. An automated chemistry analyzer (IDEXX Laboratories, Westbrook, ME, USA) was employed to estimate serum triglyceride levels; the execution process of this analyzer followed the manufacturer-provided instructions [32]. On the basis of the procedure described by Tsai et al. [25], Triton X-100 solution was used to extract hepatic triglycerides from homogenized liver samples. The extracted samples were solubilized by twice slowly heating to 90 °C for 5 min and subsequently cooling to room temperature. Insoluble material was removed from the samples through centrifugation, and colorimetric assays were performed on the collected supernatant to examine triglyceride level (BioVision, Milpitas, CA, USA).

### 2.6. Hepatic and Renal Catalase, Glutathione Peroxidase, and Superoxide Dismutase Levels

To determine hepatic and renal superoxide dismutase (SOD), glutathione peroxidase (GPx), and catalase (CAT) activities, the liver was perfused through the use of ice-cold saline (0.9% sodium chloride), after which it was subjected to homogenization in chilled potassium chloride (1.17%) [33]. Next, 5 min centrifugation ($800\times g$) was performed at 4 °C to derive homogenates. The supernatant derived was subjected to 20 min centrifugation ($10,500\times g$) at 4 °C to obtain postmitochondrial supernatant from the liver and renal samples. Catalase, GPx, and SOD measurements were executed through colorimetrical

kits (#K335-100, #K762-100, and #K773-100, respectively) made commercially available by BioVision. The instructions supplied with the kits were followed.

### 2.7. IPGTT

An IPGTT was performed after 63 days of treatment with vehicle or curcumin. Specifically, 1 g/kg glucose was administered to the 22-week-old mice after an overnight fast during which water was provided ad libitum. After 0, 30, 60, 90, and 120 min, blood was collected from the tail vein, and a One Touch glucose meter (LifeScan, Milpitas, CA, USA) was employed to measure the blood glucose level. Glucose tolerance was estimated according to the area under the curve (AUC) for glucose level for the first 120 min after glucose administration [25].

### 2.8. Western Blotting

After euthanization of the mice, their gastrocnemius muscles were immediately removed, coarsely minced, and homogenized. The following antibodies were employed in Western blotting [3]: fatty acid synthase (FASN), adiponectin, anti-Akt, anti-phospho-Akt (phosphothreonine 308-specific), anti-GLUT4, and anti-actin (Cell Signaling Technology, Beverly, MA, USA), along with PNPLA3 (Sigma-Aldrich, St. Louis, MO, USA). Immunoreactive signals were detected through the employment of enhanced chemiluminescence reagents (Thermo Scientific, Rockford, IL, USA), and each membrane was subsequently exposed on an X-ray film. Scion Image (National Institute of Health, Bethesda, MD, USA) was used with the films to quantify phosphorylation and protein expression.

### 2.9. Analysis of Chromium Level

The samples harvested at the end of the experiment were rinsed with saline and blotted dry before being weighed. As reported in previous publications [4,20], the chromium levels of these samples were determined. In brief, 65% nitric acid (1 mL) was employed at 100 °C to digest a 25 μL blood sample or 0.1 g tissue sample. Distilled water was then used to dilute the digests to a 5 mL volume. Single-particle inductively coupled plasma–mass spectroscopy (NexIon 350X; Perkin Elmer, Waltham, MA, USA) was employed to identify the chromium content of each digested solution. Samples were analyzed in triplicate, and the mean level (in nanograms per milliliter of blood or urine or nanograms per gram of tissue) was calculated. The analysis method yielded a 98% relative recovery rate with a 4.2% relative standard deviation (SD) at 5 ng/g chromium (n = 5). Regression analysis ($R^2$ > 0.995) was conducted on the standard 1–500 ng/g chromium curve to determine the total chromium level in each sample [6].

### 2.10. Statistical Analysis

The derived data are presented herein in the form of mean (M) ± SD. Between-group differences were determined by executing Student's $t$ test, and the significance of contingency data was evaluated by executing Fisher's exact test; $p < 0.05$ was regarded as constituting statistical significance.

## 3. Results

### 3.1. Curcumin Influences Food and Chromium Intake, Food Efficiency, and Morphometric Parameters

The mice's body weight and body weight gain were not found to significantly differ between the SD-fed curcumin-treated group and control group in the preliminary studies (Figure S1). Thus, this study created a mouse model of obesity through an HFD lasting 10 weeks. The curcumin group received curcumin treatment for 10 weeks and had lower levels of all morphometric parameters relative to the control group (Figure 1). The differences in diet caused disparities in the metabolic parameters and chromium intake of the mice. Following the six-week curcumin treatment, the curcumin-group mice weighed 14% less than the control-group mice ($p < 0.01$; Figure 1a). Additionally, the curcumin group was

noted to achieve a 54% (Figure 1b) and 53% (Figure 1c) reduction in body weight change and weekly body weight gain. The curcumin-group mice consumed 25% less food on a weekly basis (Figure 1d), had a 23% lower weekly chromium intake (Figure 1e), and had 40% lower daily food efficiency than did the control-group mice (Figure 1f). The finding that the curcumin-treated mice gained less weight was attributed to their lower food intake. Furthermore, the curcumin group had lower body weight within 10 weeks of the curcumin treatment, indicating less weight gain relative to the control group.

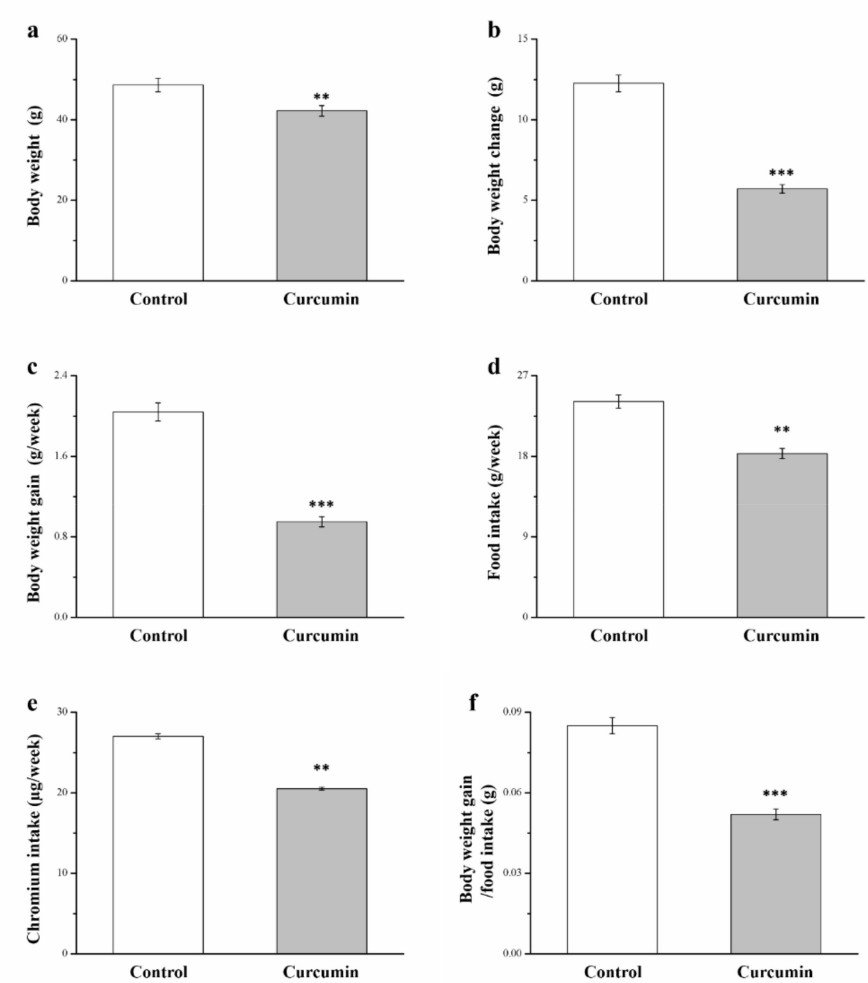

**Figure 1.** Body (**a**) weight, (**b**) weight change, and (**c**) weight gain; weekly (**d**) food and (**e**) chromium intake (per mouse); and (**f**) daily food efficiency of the curcumin and control groups. All values are presented herein as M ± SD; n = 10 per group. ** $p < 0.01$ and *** $p < 0.001$.

### 3.2. Curcumin Affects Liver Fat Infiltration and Adipocyte Size

The between-group differences in fat accumulation in liver and adipose tissues were analyzed. The results of the morphometric analysis after H&E staining revealed that relative to those in the control group, the mice given curcumin for 10 weeks had less fatty liver and smaller fat cells in RWAT and EWAT (Figure 2a). Additionally, the curcumin group was noted to have a 36% lower average liver fat infiltration score (Figure 2b). The fat cells in RWAT and EWAT were also 15% (Figure 2c) and 20% (Figure 2d) smaller in the curcumin group. In sum, this study demonstrated that the curcumin-treated mice with obesity had less liver fat infiltration and fat cell hypertrophy, even when on an HFD.

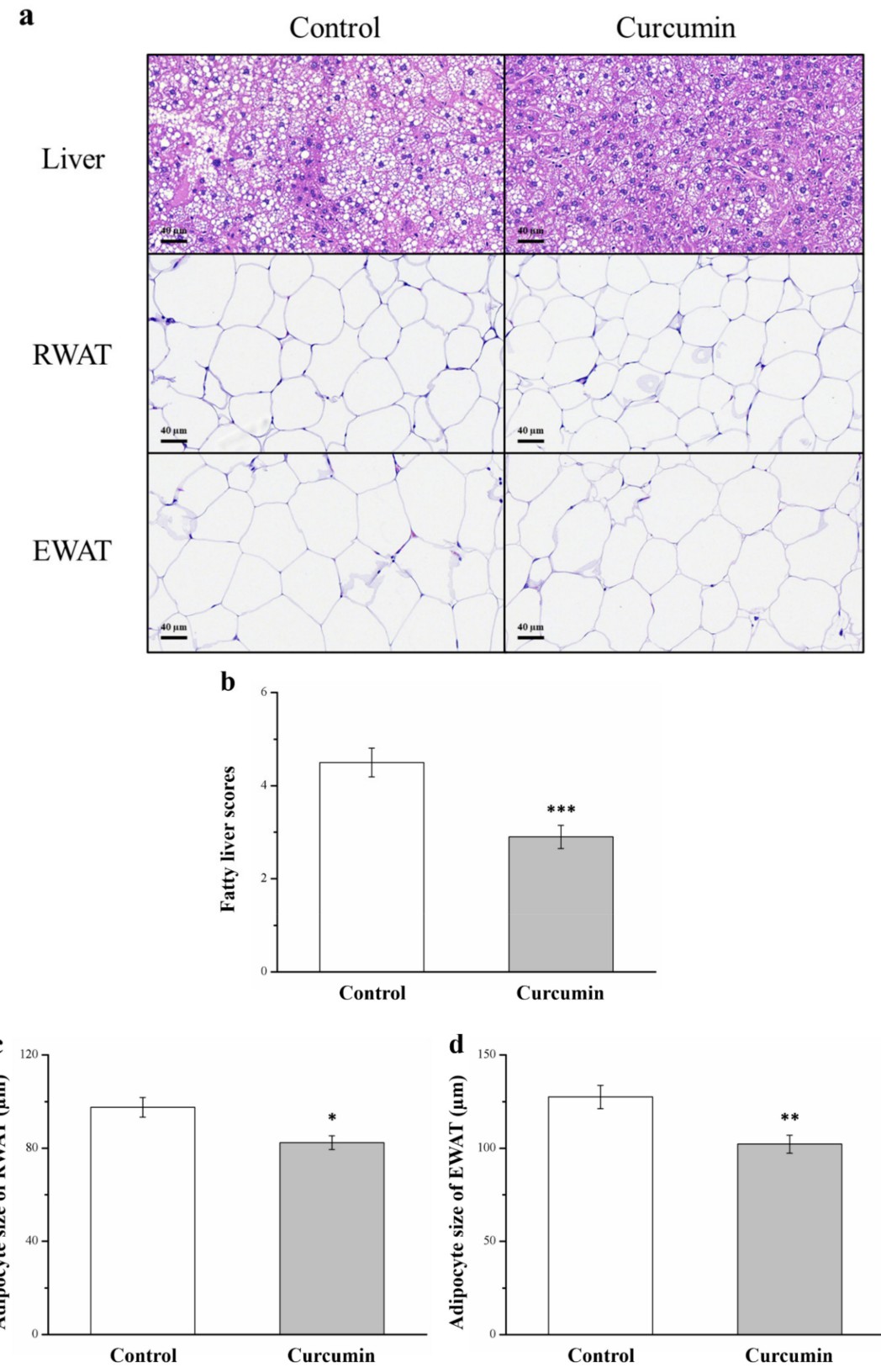

**Figure 2.** (**a**) Photomicrographs of hematoxylin and eosin (H&E)-stained liver tissue, retroperitoneal white adipose tissue (RWAT), and epididymal white adipose tissue (EWAT) sections; (**b**) liver fat infiltration score; and adipocyte cellularity of (**c**) RWAT and (**d**) EWAT in the curcumin and control groups. Values are presented herein as M $\pm$ SD; n = 10 per group. All photomicrographs were employed in histological analysis (magnification: 200$\times$). * $p < 0.05$, ** $p < 0.01$, and *** $p < 0.001$.

### 3.3. Curcumin Affects Serum and Hepatic Levels of Triglycerides and Hepatic Expression of FASN, PNPLA3, and Adiponectin

Assessments were performed to determine if the observed differences in liver fat infiltration were related to differing serum or hepatic triglyceride levels. After 10 weeks of curcumin treatment, body composition significantly differed between the treated and control mice in terms of serum (Figure 3a) and hepatic (Figure 3b) triglyceride levels. The curcumin group had 17% and 18% higher serum (Figure 3a) and hepatic (Figure 3b) triglyceride levels than did the control group. To identify curcumin's molecular effects on a fatty liver (Figure 3c), the levels of fatty liver marker expression (i.e., the effect of curcumin on triglyceride synthesis and lipid homeostasis regulation) were measured [2,25,26]. The Western blotting of liver tissues suggested that the curcumin treatment reduced FASN expression by 67% (Figure 3d). Additionally, the curcumin group was determined to have a 56% reduction in the expression of patatin-like phospholipid domain containing protein (PNPLA3) (Figure 3e). The curcumin group had 58% higher adiponectin expression in the liver (Figure 3f). Thus, the curcumin-treated obese mice experienced a greater decrease in the levels of serum triglycerides, fat-synthesis-related proteins such as FASN and PNPLA3, and hepatic triglycerides, but they experienced significantly higher adiponectin levels.

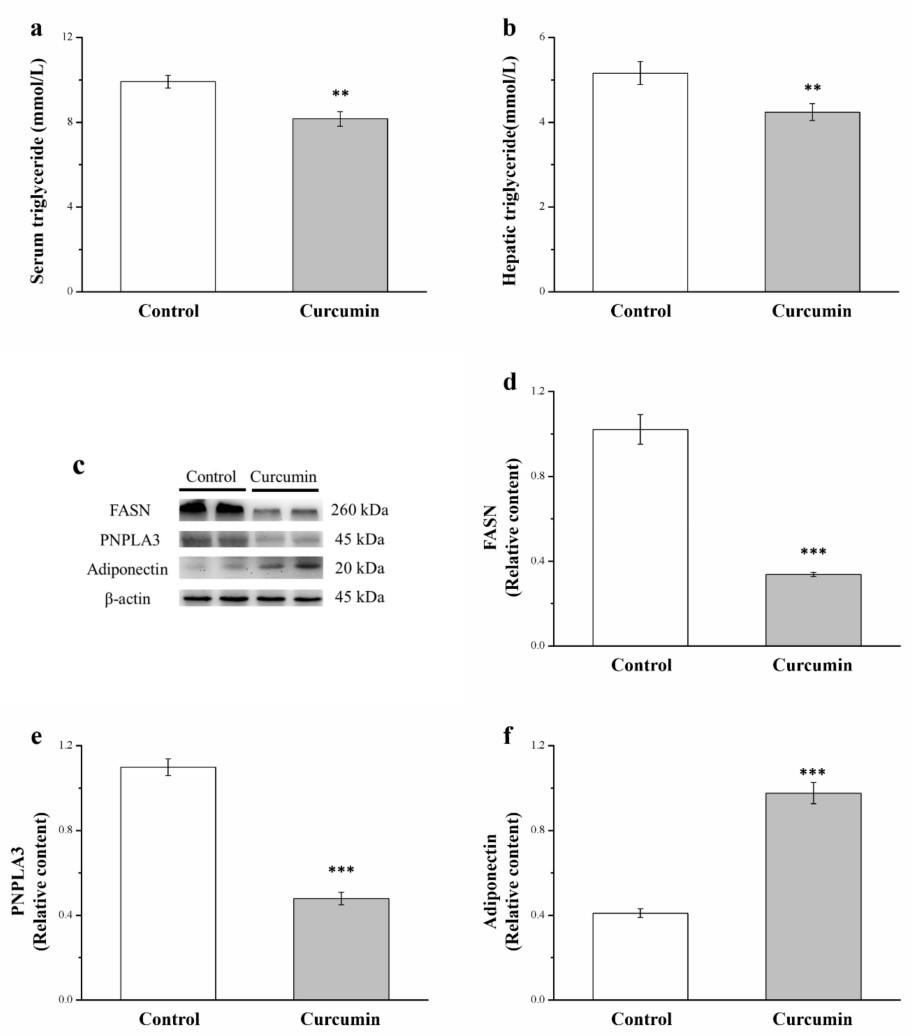

**Figure 3.** Levels of (**a**) serum and (**b**) hepatic triglycerides; (**c**) Western blotting for FASN, PNPLA3, and adiponectin; and (**d**) FASN, (**e**) PNPLA3, and (**f**) adiponectin expression in the curcumin and control groups. All values are expressed as M ± SD; n = 10 per group. ** $p < 0.01$ and *** $p < 0.001$.

### 3.4. Curcumin Affects Serum Levels of ALT, AST, and Hepatic Antioxidant Enzymes

The curcumin group's serum ALT (Figure 4a) and AST (Figure 4b) levels were determined to be 22% and 24% lower than the levels identified in the control group. ALT and AST are hepatic function markers. The pathogenesis of NAFLD is linked to the liver containing a reduced quantity of antioxidant enzymes, and this decreased antioxidant activity may cause steatosis and steatohepatitis [34]. The curcumin group exhibited 49%, 62%, and 45% higher levels of the antioxidant biomarkers CAT (Figure 4c), GPx (Figure 4d), and SOD (Figure 4e), respectively. Therefore, curcumin reduced the levels of hepatic function indexes, including AST and ALT, and increased the levels of the hepatic antioxidant enzymes SOD, GPx, and CAT in obese mice.

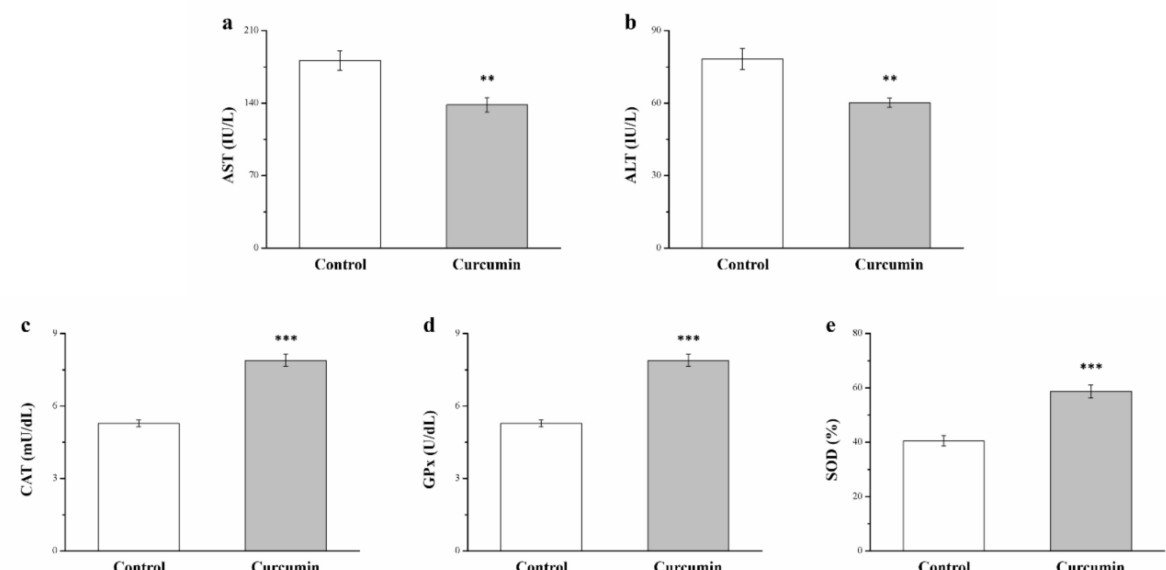

**Figure 4.** Levels of serum (**a**) aspartate aminotransferase (AST) and (**b**) alanine aminotransferase (ALT), and levels of hepatic (**c**) catalase (CAT), (**d**) glutathione peroxidase (GPx), and (**e**) superoxide dismutase (SOD) activity in the curcumin and control groups. All values are presented herein as M $\pm$ SD; n = 10 per group. ** $p < 0.01$ and *** $p < 0.001$.

### 3.5. Curcumin Affects Serum Insulin Level and Glucose Intolerance

To determine whether curcumin affected glucose homeostasis, blood glucose was measured five times in the 120 min after glucose administration (at 0, 30, 60, 90, and 120 min). The curcumin group's subsequent increases in blood glucose were identified to be greater than the levels identified in the control group (Figure 5a). The AUC of glucose level for this post-glucose-administration period was 21% lower for the curcumin group (Figure 5b). This study applied the following glucose intolerance criterion at 120 min after glucose administration: blood glucose level > 9 mmol/L. The curcumin-group animals were discovered to be less glucose intolerant (Figure 5c). However, they also exhibited significantly higher serum insulin levels (Figure 5d). Thus, even in a state of more severe insulinemia, the curcumin-treated mice exhibited predominant glucose homeostasis.

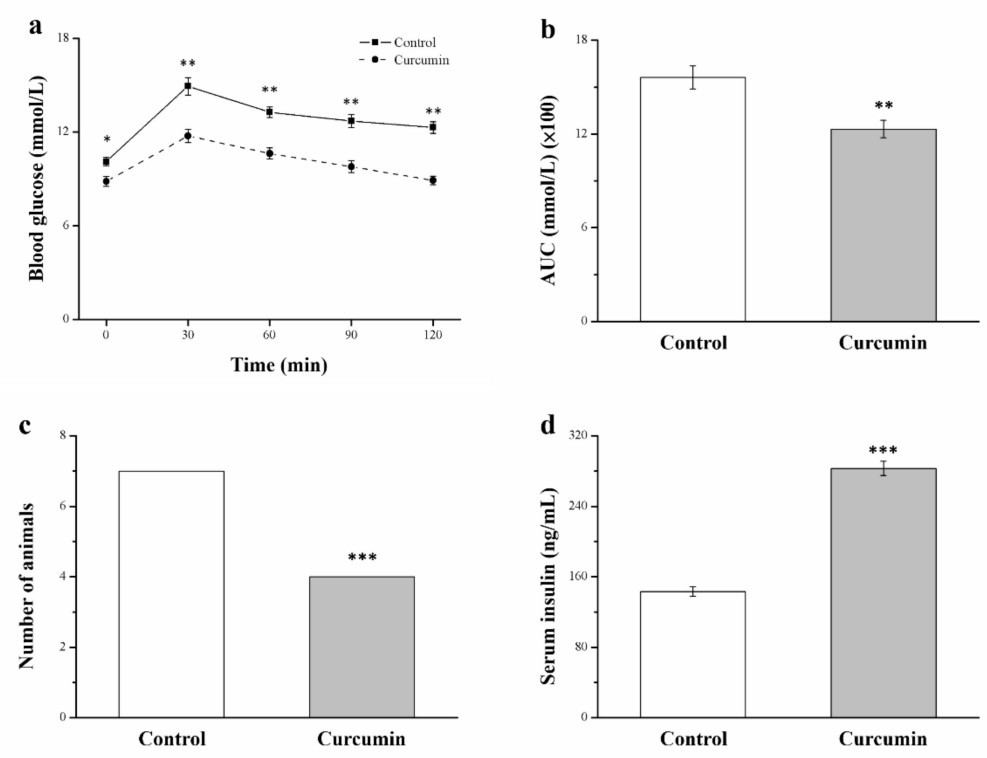

**Figure 5.** (**a**) Tolerance to 1 g/kg glucose, (**b**) area under the curve for the 120 min period following glucose administration, (**c**) number of mice with glucose intolerance (Fisher's exact test), and (**d**) levels of serum insulin in the curcumin and control groups. All values are expressed as M ± SD; n = 10 per group. * $p < 0.05$, ** $p < 0.01$, and *** $p < 0.001$.

### 3.6. Curcumin Affects Akt and GLUT4 Expression and Insulin Sensitivity

The homeostasis model assessment-insulin resistance (HOMA-IR) [20] and insulin sensitivity indices [27] were employed as measures of insulin sensitivity, indicating curcumin's effect on glucose homeostasis. The HOMA-IR index for the curcumin group was 1.4 times that for the control group (Figure 6a). The curcumin-treated mice also had a 30% lower insulin sensitivity index (Figure 6b).

At the end of this study, Akt phosphorylation and glucose transporter 4 (GLUT4) expression in muscle tissue were evaluated to reveal the mechanisms underlying the curcumin group's elevated glucose transport (Figure 6c). The curcumin group was discovered to have significantly higher phospho-Akt levels (Figure 6d); a high phospho-Akt level reduces the pathological characteristic of DM [26]. The curcumin-treated mice also exhibited significantly higher GLUT4 expression (Figure 6e). Finally, the insulin sensitivity index derived for the curcumin group was noted to exceed the control group's index, possibly because of insulin signaling enhancement caused by greater expression of GLUT4 and Akt phosphorylation.

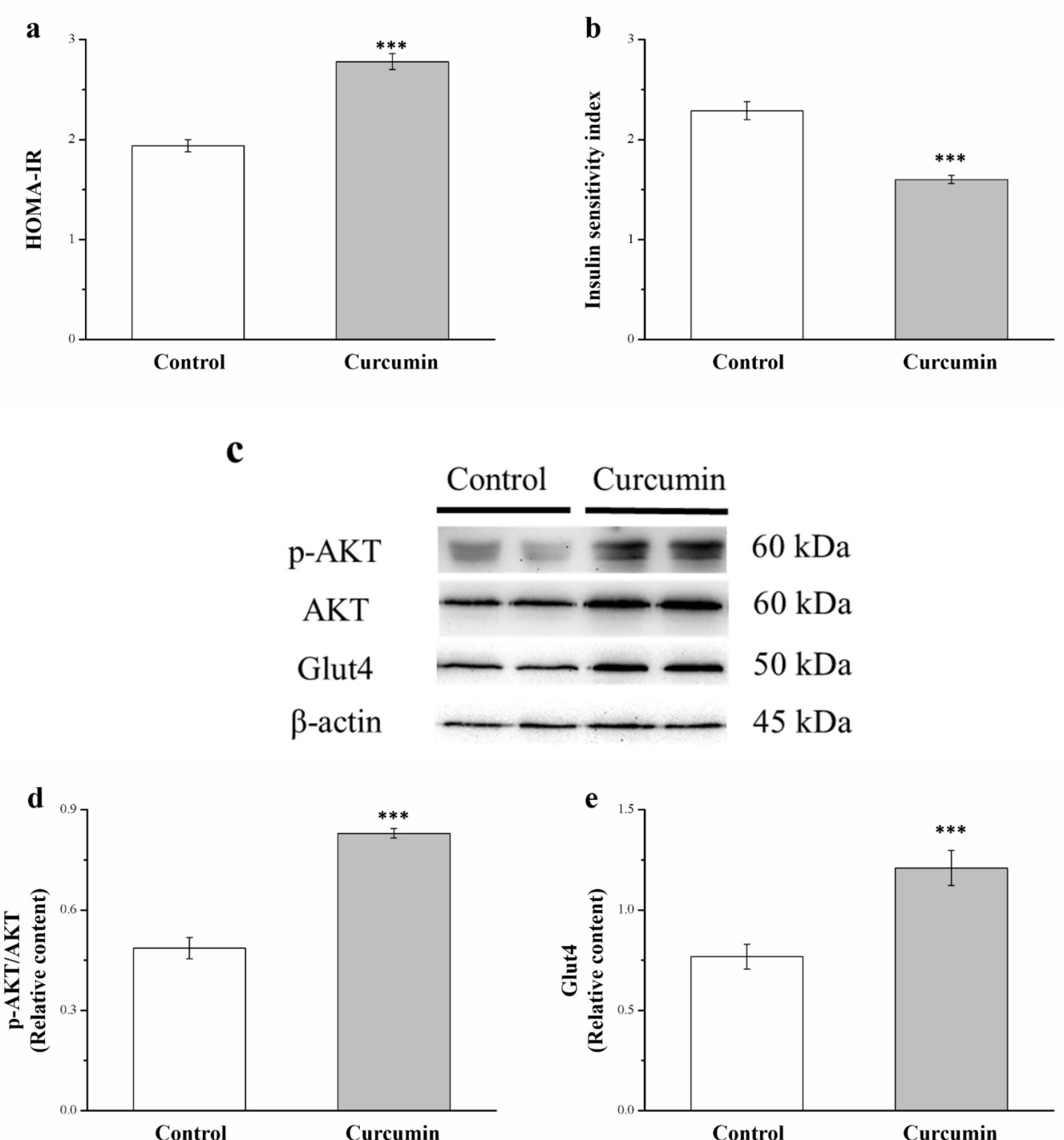

**Figure 6.** (**a**) Homeostatic model assessment of insulin resistance (HOMA-IR) index, (**b**) insulin sensitivity index, and (**c**–**e**) gastrocnemius muscle GLUT4 and phospho-Akt expression levels in the curcumin and control groups. All values are presented herein as M ± SD; n = 10 per group. *** $p < 0.001$.

### 3.7. Curcumin Changes Chromium Level in Tissues

Trivalent chromium may be crucial to glucose homeostasis, potentiating the effects of insulin. The chromium content of various harvested tissues was determined to determine whether the curcumin group, which had HFD-induced obesity and glucose intolerance, had differing chromium levels to the control group (Table 1). Bone chromium levels were approximately 33% lower in the curcumin group. The chromium levels in blood, liver, fat pads, and muscle exhibited a contrasting trend, however; on average, they were 1.43-,

1.47-, 1.28-, and 1.43-fold higher in the curcumin group. In sum, the results indicated that curcumin altered the chromium distribution, resulting in significantly higher muscle, liver, blood, and fat pad levels but significantly lower bone levels.

**Table 1.** Chromium content of organs and tissues from the curcumin and control groups.

| Variable | Control | Curcumin |
| --- | --- | --- |
| Blood (ng/mL) | 90 ± 7 | 128 ± 8 *** |
| Bone (ng/g) | 357 ± 12 | 238 ± 10 *** |
| Liver (ng/g) | 116 ± 2 | 171 ± 6 *** |
| Epididymal fat pads (ng/g) | 57 ± 8 | 73 ± 6 ** |
| Muscle (ng/g) | 102 ± 7 | 145 ± 9 *** |

All values are expressed as M ± SD; n = 10 for both groups. ** $p < 0.01$, and *** $p < 0.001$.

### 3.8. Curcumin Changes the Chromium Level in the Kidneys and Urine

Next, this study determined whether the changes in chromium levels were associated with altered urinary excretion of chromium. The results revealed that relative to those in the control group, the curcumin-treated mice with induced obesity had markedly lower chromium concentrations in their kidneys and urine (42% and 30% lower, respectively; Figure 7a,b). Thus, curcumin treatment reduced chromium loss through the kidneys and urine in obese mice.

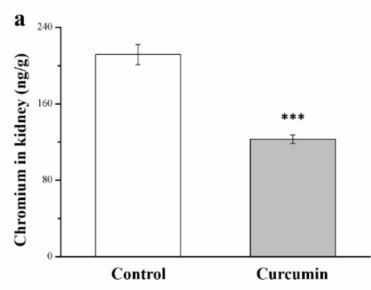 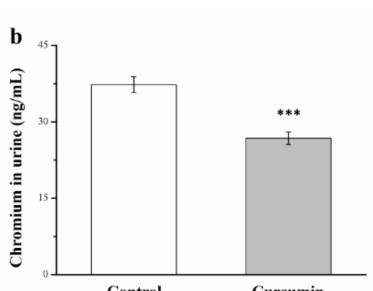

**Figure 7.** Chromium levels in (**a**) kidneys and (**b**) urine in the curcumin and control groups. All values are presented herein as M ± SD; n = 10 per group. *** $p < 0.001$.

### 3.9. Curcumin Reduces Renal Injury, Serum Blood Urea Nitrogen Level, and Creatinine Level but Increases Antioxidant Enzyme Concentrations in the Kidneys

Evidence indicates that renal injury can be decelerated through improvements in hyperlipidemia and DM [25,26]. Thus, this study examined whether curcumin reduced the amount of renal damage. As indicated by H&E staining results, the curcumin group had less glomerulonephritis with inflammatory cell infiltration than did the control group (Figure 8a). The serum blood urea nitrogen (BUN) and creatinine levels of the curcumin group were 2.3% and 2.8% lower ($p < 0.01$; Figure 8b,c, respectively). Improved renal injury is linked to increased antioxidant enzymes in the kidney; these enzymes may slow renal nephropathy and improve the kidneys' functioning [33]. The curcumin group exhibited 1.6-, 1.5-, and 1.3-fold higher catalase (Figure 8d), GPx (Figure 8e), and SOD (Figure 8f) activities compared with the control group. Therefore, the obese mice given curcumin treatment had less severe renal injury, lower levels of BUN and creatinine, and more numerous renal antioxidant enzymes—catalase, GPx, and SOD.

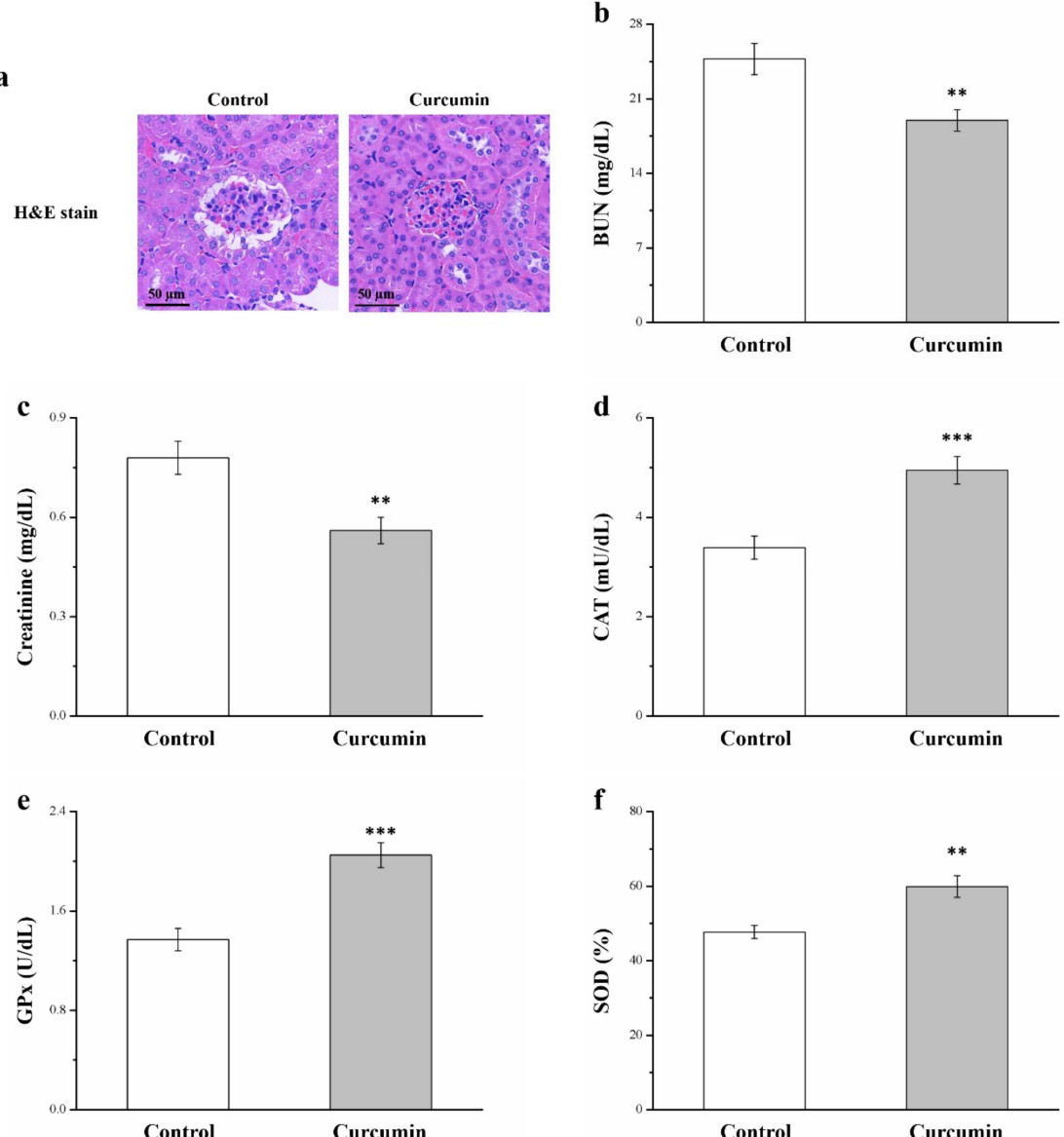

**Figure 8.** (**a**) H&E-stained kidney images (magnification, 200×); serum levels of (**b**) blood urea nitrogen (BUN) and (**c**) creatinine; and renal activity of (**d**) CAT, (**e**) GPx, and (**f**) SOD in the curcumin and control groups. All values are presented herein as M ± SD; n = 10 per group. ** $p < 0.01$ and *** $p < 0.001$.

## 4. Discussion

This study obtained the following findings. (1) Curcumin was discovered to significantly decrease the caloric intake and body weight of mice with obesity caused by an HFD. (2) Curcumin also significantly decreased the fatty liver score, adipocyte size, and renal damage severity of these mice. (3) The curcumin group's glucose tolerance and AUC for glucose level were significantly lower than those derived for the control group also fed an HFD; similar results were noted for serum insulin levels. (4) In the curcumin-treated HFD group, blood and bone chromium levels were significantly lower, but liver, fat pad, and muscle chromium levels were significant higher. (5) The curcumin-treated mice on an HFD exhibited significantly higher insulin sensitivity and lower insulin resistance than did the HFD control group; these results were similar to those obtained for the attenuation of Akt and GLUT4 expression, both of which are involved in insulin signaling.

In agreement with existing results [3,35], this study found that the HFD significantly promoted obesity development and increased body weight and body and organ fat accu-

mulation. Curcumin decreased the body weight of HFD-fed mice, and this was attributed to decreased food intake. These results indicate that curcumin prevents high food intake and weight gain and decreases daily food efficiency. In addition, the decrease in body fat was greater in the animals given curcumin in addition to an HFD than in those fed only an HFD; this was possibly because of the decreases in the average RWAT and EWAT adipocyte sizes and in liver fat infiltration. Curcumin can decrease the size of adipocytes that have become large through an HFD, possibly because of its attenuation of adipocyte hypertrophy. These results corroborate those of previous research [4], wherein a decrease in fat pad mass was attributed to a decline in adipocyte enlargement owing to fat storage (adipocyte hypertrophy) and adipocyte formation from precursor cells (adipocyte differentiation).

The curcumin group's weekly food intake was noted to be lower than that identified for the control group. Those administered curcumin had a smaller appetite. A similar observation was made for patients with obesity and NAFLD who were receiving curcumin supplementation [36]. The present study's curcumin group's lower serum ALT and AST levels indicate that curcumin reduced the degree to which lipids infiltrated the liver. In rats given a diet rich in lipids and then treated with streptozotocin, curcumin regulated serum ALT and AST by raising the quantity of liver antioxidant enzymes, such as SOD [37]. In addition to increasing fatty liver scores, curcumin was linked to adipogenesis in the liver, as estimated on the basis of associated proteins, including the expression of FASN, PNPLA3, and adiponectin [34]. Curcumin treatment lowered fatty liver scores in obese mice with lower FASN and PNPLA3 activation and higher adiponectin expression. However, limited mouse and human research has been conducted into how curcumin affects appetite. In one report on goldfish, the degree to which curcumin treatment reduced appetite was affected by the vagal afferent and then the corticotropin-releasing hormone (CRH)/CRH receptor pathway [38]. In addition, the present study's curcumin group's serum leptin levels were determined to be significantly lower than the levels identified for the control group (Figure S2a). A previous study compared serum leptin levels between groups because leptin is an adipokine primarily synthesized in white adipose tissue and partially regulates food intake through related hypothalamus activity [3]. Leptin's physiological activity, affecting appetite and body weight homeostasis, involves the suppression of neuropeptide Y in the hypothalamus and the induction of anorexigenic factors, such as forkhead box protein, cocaine- and amphetamine-regulated transcripts, agouti-related peptide, and proopiomelanocortin [4,39]. However, despite having higher serum leptin levels, the control group of the present study did not consume less food than the curcumin group did. The leptin resistance of the control mice, which were fed the HFD, agrees with their lower serum leptin receptor levels (Figure S2b). Furthermore, central leptin resistance, which occurs because of leptin transport, signaling, and impairment of the target neural circuit, has been concluded to be a main obesity risk factor. Accordingly, the mice with HFD-induced obesity had considerably fewer leptin receptors—and increased appetite attributable to leptin resistance—and higher food intake. Evidently, curcumin improved leptin resistance, contributing to a lower caloric intake and thus preventing obesity aggravation.

Previously executed research reported that HFDs raise healthy human [40] and animal model [3,25] blood glucose levels. This study corroborates these findings by showing significantly increased blood glucose levels in mice with HFD-induced obesity. In some studies, curcumin lowered blood glucose and improved glucose tolerance by prompting glucagon-like peptide-1 (GLP-1) and insulin secretion in rats and diabetic mice [41,42]. However, short-term curcumin treatment in mice injected with dexamethasone negligibly affected the insulin concentration in plasma [43]. In the present study, obese mice administered curcumin exhibited significantly higher glucose tolerance and serum insulin levels relative to those receiving no curcumin; these improvements were accompanied by a lower blood glucose level (>9 mmol/L, the glucose intolerance criterion), which lasted for 120 min after glucose administration. The apparent inconsistency in curcumin's effects on insulin levels can probably be ascribed to differences between treatments administered

in the short versus long term. The decrease was supported by the glucose-level-related AUC, another index reflecting changes in glucose level [20,44]. These results confirm that decreases in adipose tissue mass are associated with decreased IPGTT-assessed glucose intolerance, and these effects reverse hyperglycemia in mouse models of obesity.

This study also noted that even in a hypoinsulinemic state, mice with HFD-induced obesity had impaired glucose homeostasis. Thus, progression to hypoinsulinemia may impair glucose homeostasis [25]. The curcumin-treated mice on the HFD in the present study had significantly higher insulin levels than did control mice on the HFD. However, an HFD has been discovered to promote pancreatic β-cell death and then lead to a decrease in insulin secretion [27,45]. In addition, some studies have reported that HFD-fed mice developed hypoinsulinemia [3,46]. This difference in insulin levels is likely due to a difference in the proliferation of pancreatic β-cells between HFD-fed mice without and with curcumin treatment [30,31]. The results further indicate that curcumin increased serum insulin levels; this was demonstrated as by the long-term use of curcumin causing an increased β-cell percentage in the pancreas (Figure S3). This benefit is potentially attributable to β-cell regeneration and survival during HFD feeding and concurrent curcumin treatment. The HOMA-IR index, which quantifies insulin resistance, reflects HFD-induced changes in circulating insulin and glucose levels. Generally, high adipose tissue mass and liver steatosis are risk factors for insulin resistance [36]. However, some clinical studies have indicated that daily curcumin supplementation (length ≤10 weeks; curcumin dose ≥ 1500 mg) decreased the fasting blood glucose level in type 2 DM; however, changes in glycated hemoglobin A1c (HBA1c), serum insulin level, and insulin resistance were found to be nonsignificant [47,48]. Another clinical study suggested that 300 mg of curcuminoid administered on a daily basis for 3 months resulted in reduced insulin resistance, HBA1c, and fasting blood glucose in type 2 DM [49]. In other words, curcumin treatment lasting less than 10 weeks, even when using a high dose, did not make individuals with type 2 DM less insulin resistant. Treatment length, frequency, and species probably affect the outcomes of curcumin treatment. Therefore, curcumin could improve insulin resistance, and blood glucose imbalance attenuates glucose intolerance. Furthermore, curcumin has more beneficial effects on DM when combined with an antidiabetic drug [50].

The insulin sensitivity index represents the effectiveness with which insulin modulates glucose [3]. In obese mice, reduced Akt phosphorylation leads to weakening of insulin signaling, thus impairing glucose homeostasis and insulin resistance [51]. In this study, the curcumin group had a higher insulin sensitivity index than did the control group, possibly because of the greater insulin-stimulated Akt phosphorylation noted in the curcumin-treated mice. However, J17, which is structurally similar to curcumin, markedly reduced Akt phosphorylation in a myoblast cell line (H9C2 cells) under hyperglycemia-induced inflammation conditions [52]. Additionally, curcumin effectively increased insulin signaling expression, including expression of glucose transporter 2, and the phosphorylation of insulin receptors, AKT, phosphatidylinositol-3-kinas, and insulin receptor substrate-1 in the INS-1 insulinoma cell line [53]. The rate-limiting step during the process of muscle glucose metabolism is glucose transporter 4 (GLUT4)-mediated transmembrane glucose transport. Poor basal glucose transport suggested severe insulin resistance and glucose intolerance in diabetic mice that selectively lacked GLUT4 in their muscles [34,54]. This study found that muscle GLUT4 expression was increased in the curcumin group, probably because the stimulation of muscle glucose uptake was enhanced in response to the hyperglycemia induced by the HFD and curcumin supplement [55]. Stimulation of the insulin signaling pathway may increase insulin sensitivity and help overcome hyperglycemia, which occur in parallel with hyperinsulinemia [26]. In sum, our results imply that curcumin can improve severe hyperglycemia and insulin sensitivity in mice with obesity induced by an HFD.

Chromium is obtained from diet and physiologically stored in bone. Chromium released from bone is then moved as required to regulate the uptake of glucose in such metabolic organs as the liver and in such tissues as adipose tissues and skeletal muscle. This leads to insulin signal transduction activation, thus ensuring the regulation of glucose

homeostasis [20,35]. Curcumin treatments given to HFD-fed mice considerably lowered bone chromium levels in a comparison of the present study's curcumin and control groups. The present results also indicate that chromium levels were higher in the blood, liver, muscle, and fat pads of the curcumin-treated HFD-fed mice. Thus, this study's results confirm that curcumin has a positive influence on chromium accumulation in selected tissues and has a positive effect on HFD-induced glucose intolerance [56]. A similar observation was made in another study; individuals without DM had a significantly greater increase in serum chromium concentration than did patients with DM [57]. Therefore, the observed changes in chromium levels caused by curcumin treatment could ameliorate the glycemic status of obese mice with dysglycemia. Furthermore, in insulin-sensitive tissues, curcumin promoted chromium mobilization and controlled hyperglycemia in the obese mice examined in this study.

The kidney is the main mediator of essential trace metal excretion [20]. Chromium is primarily excreted through urine, and the proximal renal tubule reabsorbs the majority of minor elements [58,59]. Hyperglycemia leads to considerable loss of chromium and decreased chromium reabsorption in type 2 DM [34]. Therefore, long-term chromium loss worsens insulin resistance in HFD-fed mice and makes them more glucose intolerant [6,20]. Our study showed increased chromium levels in metabolic tissues and blood; however, the opposite was largely observed in the kidneys and urine after curcumin treatment. Thus, curcumin can reduce the large-scale movement of chromium from bone into the kidneys and decrease HFD-induced chromium loss through urine, further enhancing chromium redistribution or mobilization and preventing the aggravation of hyperglycemia in HFD-fed mice. The essential metal can help glucose-intolerant obese mice overcome hyperglycemia with curcumin treatment. Hyperglycemia can result in renal damage, decreased trace metal reabsorption, and increased urinary excretion [34]. This finding is supported by increases in the renal function indexes (BUN and creatinine) of HFD-fed hyperglycemic mice, although the results differed for the HFD-fed mice treated with curcumin. Similarly, one study showed that long-term curcumin treatment alleviated renal damage caused by chronic kidney diseases related to systemic dysfunctions, such as chronic inflammation [60]. The study also found that curcumin treatment resulted in fewer HFD-induced injury lesions in mice, in addition to glomerulonephritis. For the curcumin-treated mice, the derived findings pertaining to decreased urinary chromium and positive total chromium balance are probably related to the curcumin-induced reduction of renal injury severity. Decreased chromium loss through urine led to more favorable glucose tolerance. Thus, the results support the idea that curcumin can ameliorate hyperglycemia development through the reduction of the amount of chromium lost and prevention of kidney damage.

Metabolic syndrome or DM can cause kidney damage, which is attributable to the adverse effects of hyperglycemia [45]. Renal glomeruli, renal tubules, and vessels are harmed by hyperglycemia, and this damage can result in kidney-inflammation-associated diabetic nephropathy [61]. This study found that HFD-fed obese mice exhibited glomerulonephritis. Hyperglycemia also decreased the quantity of antioxidant enzymes, which leads to and worsens diabetic nephropathy [62]. The present findings also indicate that curcumin markedly increased renal CAT, GPx, and SOD activities. These increases in antioxidant enzyme activity can reduce renal morphological damage and the histopathological markers of inflammation [63]. Moreover, immunohistochemical staining revealed lower levels of such renal inflammatory cytokines as TNF-$\alpha$ in the curcumin-treated HFD-fed mice (Figure S4). Additionally, curcumin treatment was discovered to result in a decreased risk of obesity-related kidney diseases, such as abnormal lipid metabolism and chronic inflammation, which contribute to kidney injury [60,63]. Hence, curcumin reduced inflammation and oxidative stress by upregulating antioxidant enzyme activity, thus exhibiting salutary effects against HFD-induced chronic kidney disease (CKD).

This study showed that in HFD-fed mice, curcumin had significant positive effects on hyperglycemia—particularly the majority of its molecular, histopathological, physiological, and biochemical actions. Furthermore, in HFD-fed mice with glucose intolerance and

obesity, curcumin treatment effectively decelerated the development of insulin resistance. Curcumin also protected against HFD-induced obesity, weight gain, fatty liver, and increased serum insulin level. Moreover, curcumin decreased the amount of chromium lost through urine and lowered the severity of renal injury by fortifying the renal oxidative defense, thereby reducing renal inflammation. The curcumin-treated obese mice were noted to have significantly higher blood chromium, SOD, and HbA1c levels (Figure S5) relative to those levels noted in the controls. Chromium was discovered to be strongly associated with SOD activity in relation to HbA1c [64–66]; therefore, the present study provides strong evidence that these factors can be employed to indicate progressive complications in type 2 DM by curcumin administration. In sum, curcumin slows the development of diabetes by enhancing insulin sensitivity and reducing hepatic lipid accumulation and renal injury in HFD-fed mice, as demonstrated by this study.

## 5. Conclusions

In this study, an HFD was used to establish an obesity and hyperglycemia mouse model, which was utilized to investigate the pathogenesis of type 2 DM with insulin resistance and glucose intolerance in animals and humans. The results reveal that relative to the control group of obese mice, the curcumin group ate less and had lower body weight, smaller weight gain, less fat infiltration of the liver, smaller fat pad adipocytes, and impaired glucose homeostasis (due to reduced insulin sensitivity). Furthermore, curcumin reduced serum ALT, AST, and hepatic triglyceride levels by decreasing the hepatic lipogenesis of FASN and PNPLA3 expression; it also increased the quantity of hepatic antioxidant enzymes that protect against HFD-induced oxidative stress. These alterations in glucose metabolism were attributable to increased Akt phosphorylation and GLUT4 activity in skeletal muscle. Curcumin treatment in obese mice caused tissue chromium redistribution, which contributed to the control of hyperglycemia. This was reflected in the accumulation of chromium in glucose-metabolic tissue such as muscle, the liver, and fat pads (after release from the physiological reserve pool) and reduced chromium loss through urine. Curcumin also reduced the amount of HFD-induced renal damage, which was indicated by the presence of more antioxidant enzymes and less neuroinflammation. The underlying mechanism may involve the fortification of antioxidative defenses in the liver and kidneys. Thus, even after an HFD induces hyperglycemia, curcumin treatment may be able to facilitate glucose homeostasis. The therapeutic usefulness of curcumin in patients with chronic liver and kidney diseases but without hyperglycemia should be investigated.

**Supplementary Materials:** The following are available online at https://www.mdpi.com/article/10.3390/pr9071132/s1, Figure S1. Body (a) weight and (b) weight gain in the curcumin and control groups both fed an SD for a 10-week period. Figure S2. Serum (a) leptin and (b) soluble leptin receptor levels in the curcumin and control groups. Figure S3. (a) H&E and immunohistochemical (IHC) staining showing islet morphology and (b) β-cell percentage of the curcumin and control groups. Figure S4. (a) H&E staining (magnification, 200×) showing renal morphology and (b) kidney levels of TNF-α in the curcumin and control groups. Figure S5. HbA1c levels in the curcumin and control groups. Serum leptin and soluble leptin receptor concentrations were determined with a mouse enzyme-linked immunosorbent assay kit (Zgenebio Biotech Inc., Taipei, Taiwan). In the mice's pancreatic islets, primary antibodies against insulin were employed for IHC staining for insulin (Bioss, Woburn, MA, USA). In the kidneys, primary antibodies against TNF-α were employed for IHC staining for TNF-α (Merck, Billerica, MA, USA). The TAlink mouse/rabbit polymer detection system (BioTnA, Kaohsiung, Taiwan) was used to measure protein expression through IHC. An enzyme-linked immunosorbent assay kit (BioVision, Milpitas, CA, USA) was used for HbA1c measurement. The manufacturer's protocols were followed in the experiment, and the findings were read at 450 nm.

**Author Contributions:** G.-R.C. and W.-T.H. conceived the study and performed experiments; L.-S.C., C.-S.L., C.-F.W. and J.-W.L. assisted in recombinant construction; W.-L.L., T.-C.L., H.-J.L. and C.-Y.K. analyzed the data; and C.-F.L. reviewed and edited the manuscript. All authors have read and agreed to the published version of the manuscript.



**Funding:** This study was supported in part by Grant 110A3-038 from the National Chiayi University (Taiwan).

**Institutional Review Board Statement:** The review of our experimental protocol was conducted by National Chiayi University's Institu-tional Animal Care and Use Committee, who approved it under the approval No. 107029.

**Informed Consent Statement:** Not applicable.

**Data Availability Statement:** The data presented in this study are available on request from the corresponding author.

**Acknowledgments:** The authors would like to thank LiTzung Biotechnology, Kaohsiung, Taiwan, for providing pathological assistance for this study.

**Conflicts of Interest:** The authors declare no conflict of interest.

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
