# Peer review of "Curcumin Improved Glucose Intolerance, Renal Injury, and Nonalcoholic Fatty Liver Disease and Decreased Chromium Loss through Urine in Obese Mice"

_processes, doi:10.3390/pr9071132_

Round 1
Reviewer 1 Report
In this article titled "Curcumin improved glucose intolerance, nonalcoholic fatty liver disease, and renal injury and decreased chromium loss in obese mice" Chang and coauthors show how administering curcumin to mice can have beneficial effects on some biological parameters and contribute to creating a positive chromium balance.
Some clarifications are needed before publication.
Line 72: "Several animal studies and clinical trials have suggested that" ... the main effect of curcumin is the antioxidant one, see "Eudragit S100 Entrapped Liposome for Curcumin Delivery: Anti-Oxidative Effect in Caco-2 Cells", Coatings 2020, 10, 114, please add.
Line 93: "chromium supplementation reversed glucose" ... in what form? With what oxidation number? Please clarify whenever chromium supplementation is mentioned.
Figure 1: "change in body weight change", please check.
Table 1: Please check the number of significant digits and how the error is reported. For example, in 90.16 ± 7.21 it is meaningless to report the decimal digits if the error already affects the integers. It is more appropriate to report 90 ± 7.
Since animal models were used in the study, did the authors operate in compliance with the legislation on animal testing? I have not found any statements about it.
Reviewer 2 Report
Dear Editor and Authors,
Below you find some points to improve your investigation.
In the beginning, I would like to express a word of my appreciation for the work you put in conducting the research and writing the manuscript.
Your manuscript provides an interesting mouse model of obesity and hyperglycemia that can be used to study the pathogenesis of type 2 DM with glucose intolerance and insulin resistance in animals and humans. insulin resistance in animals and humans.
However, I find some vagueness points so I would like you to provide clarification.
- In the results description section, summarize the current model by the data presented and in a separate paragraph describing the correlations between all results. Is there any strong relationship between these correlations?
- Figure 2a is unreadable please improve its quality.
- The different parts in the experimental section should be better described because of the accuracy of the tests performed and the details of how the measurements were performed. Please also refer to the literature using appropriate references.
- The authors write of reduced chromium loss in obese mice. Please provide results relevant to chromium and other vital elements. So that one can see what is actually happening in the life system.
- Please show the relevant tests by liver and kidney for curcumin used, and the relevant results for vital elements in correlation to curcumin used.
Round 2
Reviewer 2 Report
Accept